# A qualitative study to explore healthcare providers' perspectives on barriers and enablers to early detection of breast and cervical cancers among women attending primary healthcare clinics in Johannesburg, South Africa

Gugulethu Tshabalala[1], Charmaine Blanchard[2,3], Keletso Mmoledi[2,3], Desiree Malope[2,4], Daniel S. O'Neil[5], Shane A. Norris[6], Maureen Joffe[2,4,5‡], Janan Janine Dietrich[1,7‡]*

1 Perinatal HIV Research Unit, Faculty of Health Sciences, University of the Witwatersrand, Johannesburg, South Africa, 2 Strengthening Oncology Services Research Unit, Faculty of Health Sciences, University of the Witwatersrand, Johannesburg, South Africa, 3 Centre for Palliative Care, Department of Medicine, Faculty of Health Sciences, University of the Witwatersrand, Johannesburg, South Africa, 4 SAMRC Developmental Pathways to Health Research Unit, Department of Paediatrics, Faculty of Health Sciences, University of Witwatersrand, Johannesburg, South Africa, 5 Yale Cancer Center, Department of Medicine, Yale School of Medicine, Yale University, New Haven, United States of America, 6 South African Medical Research Council Common Epithelial Cancer Research Centre, Tygerberg, South Africa, 7 Health Systems Research Unit, South African Medical Research Council, Bellville, South Africa; and African Social Sciences Unit of Research and Evaluation (ASSURE), division of Wits Health Consortium, Johannesburg, South Africa

‡ MJ and JJD share senior authorship on this work.
* dietrichj@phru.co.za

## Abstract

Low-and-middle income countries (LMICs) contribute approximately 70% of global cancer deaths, and the cancer incidence in these countries is rapidly increasing. Sub-Saharan African (SSA) countries, including South Africa (SA), bear some of the world's highest cancer case fatality rates, largely attributed to late diagnosis. We explored contextual enablers and barriers for early detection of breast and cervical cancers according to facility managers and clinical staff at primary healthcare clinics in the Soweto neighbourhood of Johannesburg, South Africa. We conducted qualitative in-depth interviews (IDIs) between August and November 2021 amongst 13 healthcare provider nurses and doctors as well as 9 facility managers at eight public healthcare clinics in Johannesburg. IDIs were audio-recorded, transcribed verbatim, and entered into NVIVO for framework data analysis. Analysis was stratified by healthcare provider role and identified *apriori* around the themes of barriers and facilitators for early detection and management of breast and cervical cancers. Findings were conceptualised within the socioecological model and then explored within the capability, opportunity and motivation model of behaviour (COM-B) for pathways that potentially influence the low screening provision and uptake. The findings revealed provider perceptions of insufficient South African Department of Health (SA DOH) training support and staff rotations resulting in providers lacking knowledge and skills on cancer, screening policies

**Data Availability Statement:** In accordance with the provisions of the Protection of Personal Information Act 4 of 2013, participants have to consent for their data to be shared with third parties. However, at the time of study approval and data collection, participants had not consented for their data to be shared with third parties. Therefore, to maintain participant privacy and confidentiality, the qualitative coded data datasets are not publicly available. The coded data would be accessible through email correspondence to info@phru.co.za.

**Funding:** This study was financially supported through a grant by Bristol Myers Squibb Foundation-Secure the Future and BMSF-Multi-Sector Coalition in Implementation Research in Global Oncology (CIRGO). through this grant mechanism GT and JD received partial salary support. The specific roles of these authors are articulated in the 'author contributions' section. This work was also financially supported by the South African Medical Research Council (SAMRC) through its Division of Research Capacity Development under the SAMRC Early Investigators Programme (via funding received from the South African National Treasury). There are no relevant grant or award numbers to declare related to this funding. The funders had no role in study design, data collection and analysis, decision to publish, or preparation of the manuscript.

**Competing interests:** The authors have read the journal's policy and have the following competing interests: This study was financially supported through a grant by Bristol Myers Squibb Foundation-Secure the Future and BMSF-Multi-Sector Coalition in Implementation Research in Global Oncology (CIRGO). Through this grant mechanism GT and JD received partial salary support. This does not alter our adherence to PLOS ONE policies on sharing data and materials. There are no patents, products in development or marketed products associated with this research to declare.

and techniques. This coupled with provider perceptions of poor patient cancer and screening knowledge revealed low capacity for cancer screening. Providers also perceived opportunity for cancer screening to be undermined by the limited screening services mandated by the SA DOH, insufficient providers, inadequate facilities, supplies and barriers to accessing laboratory results. Providers perceived women to prefer to self-medicate and consult with traditional healers and access primary care for curative services only. These findings compound the low opportunity to provide and demand cancer screening services. And because the National SA Health Department is perceived by providers not to prioritize cancer nor involve primary care stakeholders in policy and performance indicator development, overworked, unwelcoming providers have little motivation to learn screening skills and provide screening services. Providers reported that patients preferred to go elsewhere and that women perceived cervical cancer screening as painful. These perceptions must be confirmed for veracity among policy and patient stakeholders. Nevertheless, cost-effective interventions can be implemented to address these perceived barriers including multistakeholder education, mobile and tent screening facilities and using existing community fieldworkers and NGO partners in providing screening services. Our results revealed provider perspectives of complex barriers to the early detection and management of breast and cervical cancers in primary health clinic settings in Greater Soweto. These barriers together appear potentially to produce compounding effects, and therefore there is a need to research the cumulative impact but also engage with stakeholder groups to verify findings and create awareness. Additionally, opportunities do exist to intervene across the cancer care continuum in South Africa to address these barriers by improving the quality and volume of provider cancer screening services, and in turn, increasing the community demand and uptake for these services.

## Introduction

One in five deaths globally are attributed to cancers [1]. Low-and-middle income countries (LMICs) contribute an estimated 70% of global cancer deaths, and sub-Saharan Africa (SSA), including South Africa (SA), bears high cancer case fatality rates amidst rapidly increasing cancer incidence [2, 3]. These poor outcomes are partially caused by late-stage at diagnosis caused by complex community, patient, provider, health system, and environmental factors [4–9].

Most African countries have developed national cancer control plans aligned with the WHO Sustainable Development Goals to guide cost-effective interventions to address late-stage cancer at diagnosis [10]. In 2017, the South African Health Department released national policies [11] and guidelines for the prevention, screening, and early detection of three prevalent cancers: breast [12], cervical [13] and prostate cancer [14].

With respect to breast cancer, early detection in high-income countries (HICs) is achieved through population screening mammography. SA and other (LMICs) do not have the skilled human resources, infrastructure, nor financial resources to implement population mammography screening in the public health system. Investigators from India have demonstrated reduction in breast cancer mortality using clinical breast examination (CBE) as a screening technique provided in community settings using trained fieldworkers [15]. SA National Department of Health current policy and guidelines state that women 40 years of age and above should undergo CBE screening when attending primary health clinics [12]. However,

these national breast cancer early detection guidelines are not formally implemented in the primary health system, and indicators for performance reporting have not been developed [16, 17].

Cervical cancer incidence among women in SA is significantly higher than in HIC [2, 3] and is steadily increasing, despite the introduction of human papillomavirus (HPV) vaccination in schools in 2014. SA national guidelines mandate CC Papanicolaou test (Pap test) screening via liquid-based cytology for women living with HIV of any age every 3 years for those who screen negative, and 6 weeks postnatally for pregnant women. Education, counselling, and screening are meant to be provided in the primary health system [13]. Woman who screen positive for pre-cervical cancer lesions are treated with cryotherapy and large loop excision of the transformation zone (LLETZ). In practice, HPV testing is not routinely implemented. Screening is currently around 13% in SA and other parts of SSA and associated with educational level, age, HIV status, contraceptive use, perceived susceptibility, and awareness about screening locations [18]. In SA, inadequate performance monitoring requires reporting of screening volumes but not precancerous lesion treatment [13, 19]. We explored provider perspectives on contextual enablers and barriers for guideline-concordant implementation of clinical breast examination and cervical cancer screening and precancerous lesion treatment for eligible women attending primary healthcare clinics in the urban poor neighbourhood of Soweto in Johannesburg, South Africa.

## Methods

### Study setting and design

This study was conducted in the districts of Soweto and Orange Farm. Soweto has 22 primary healthcare clinics, 6 community health care centres with limited imaging and surgical services, and two public hospitals, the tertiary-level Chris Hani Baragwanath Academic Hospital (CHBAH) and the regional, secondary-level Bheki Mlangeni District Hospital. The primary healthcare clinics are well located across Soweto to ensure proximity of 2km or less to most households within the community. Peri-urban Orange Farm, located 32 km South of Soweto, has around ten primary healthcare clinics, and one community health care centre. CHBAH diagnoses and treats cancer patients referred from Soweto and Orange Farm primary care facilities [5]. The CHBAH breast unit diagnoses and treats around 400 newly diagnosed adults with breast cancer annually and all women who screen positive for cervical cancer in primary care facilities are referred directly to the CHBAH gynaecology clinic for further diagnostic workup and treatment with radiation and chemotherapy [13]. From August to November 2021, we conducted in-depth interviews (IDIs) on screening and early detection of breast and cervical cancer with primary care facility nurses, doctors, and managers from eight primary healthcare clinics, four each in Soweto and Orange Farm.

### Ethics statement

This study was approved by the Human Research Ethics Committee (Medical) of the University of the Witwatersrand, Johannesburg, South Africa and the Research Committee of the Johannesburg Health District. Participants provided written informed consent prior to participation.

### Participant sampling and recruitment

A purposive sampling strategy was used to recruit healthcare providers (I.e., nurses and doctors) and facility managers. Participants were eligible if they had at least one year's experience

in common non-communicable diseases (NCDs) including cancer management. A researcher contacted facility managers directly at the clinics and interested managers were invited to participate in depth interviews (IDIs). Healthcare providers were recruited through a list of contact details provided by the respective facility manager or the manager directly set up interview appointments with eligible and interested healthcare providers for the researchers.

## Data collection procedures

A total of 22 one on one IDIs (13 with healthcare providers and 9 with facility managers) were conducted by the lead author, an experienced and multi-lingual, female qualitative research interviewer. The 21 IDIs conducted in person were performed in a private room at the respective primary healthcare clinics; one IDI was conducted telephonically. The interviews lasted on average 90 minutes and were conducted in the participants preferred local language and/or English. With no new codes emerging in the final IDI, data saturation was achieved. Data saturation is reached at the point where no new data is collected with each subsequent interview [20, 21].

## Measures

All participants completed a short socio-demographic questionnaire prior to participating in the interview. The semi-structured interview guides were developed from literature reviews of enablers and barriers for early detection of breast and cervical cancers (Table 1). The

**Table 1. Semi structured interview guide questions for health care providers and clinic managers.**

| **Health Care Providers** |
| --- |
| **Breast Cancer** |
| 1. What in your opinion makes it difficult or challenging for women to come with early breast symptoms to your clinic? |
| 2. What are the challenges or barriers you and your colleagues experience in providing routine clinical breast examination in your clinic? (Probe: *Probe*: healthcare setting problems, patient behaviour, perceptions, resources) |
| 3. What in your experience or opinion are the enablers to early detection of breast cancer? |
| **Cervical Cancer** |
| 1. What in your opinion makes it difficult or challenging for women to attend cervical screening in primary care clinics? |
| 2. What are the challenges or barriers you and your colleagues experience in providing routine cervical screening in your clinic? (Probe: *Probe*: healthcare setting problems, patient behaviour, perceptions, resources) |
| 3. What in your experience are the barriers to early detection of Cervical cancer? |
| 4. What in your experience are the enablers to early detection of Cervical cancer? |
| **Clinic Managers** |
| **Cervical and Breast Cancer** |
| 1. Does your clinic have the required resources (space, infrastructure and staff) to implement the cancer early detection and referral guidelines? If yes or no, please elaborate |
| 2. In your opinion what are the barriers that you and your clinical staff experience in implementing these early detection guidelines? |
| 3. In your opinion what would enable the routine implementation of these guidelines |
| 4. What would be required to be put in place to enable routine annual screening of women aged 30 and older for cervical diagnostic testing? |
| 5. What would need to be put in place to enable routine clinical breast examination for women 40 years and older |
| 6. How do staff rotations help or hinder cancer management? |
| **Recommendations** |
| 1. What support from district and National DOH structures and managers would be required to implement the early detection and referral guidelines in primary acre? |
| 2. What would be required to motivate and get buy-in from your clinical staff to routinely use the guidelines? |

interviewer used separate guides for the facility managers and healthcare providers (doctors and nurses). We focused on attitudes, awareness, and knowledge about breast and cervical cancers and existing clinic early detection activities; and on barriers and enablers for implementing national screening and early detection guidelines. We also asked providers and managers about perceived patient-related determinants. The facility managers interview guide also included their perceptions of the knowledge and ability of their doctors and nurses to detect common cancers, and their perceptions of impacts of regular staff rotations on pre- and early-stage cancer detection practices.

## Data analysis

All IDI's were audio-recorded, transcribed verbatim, translated into English, and then the audio-recording was verified with the transcription. Data analysis was led by GT who had an honours degree and JJD who had a doctoral degree. Data analyses was conducted iteratively using a thematic analysis approach [22]. First, GT and JJD entered a process of data immersion by reading and re-reading two transcripts for data familiarity. Both transcripts were coded using a line-by-line technique to assign codes to text. Codes were then categorized into themes and subthemes. Thereafter, GT and JJD developed a codebook *apriori* in Excel using two coded transcripts and the interview guides. The codebook was shared with the study team for review and input. Thereafter the codebook was entered into NVIVO 12 plus, and all transcripts were then uploaded and coded. Following the coding process, data were stratified by healthcare provider and facility managers and identified *apriori* around barriers and facilitators for early detection and management of breast and cervical cancers. All coded data were then reviewed and categorized iteratively by two members of the research team.

## Dissemination workshop

To achieve trustworthiness, we hosted a workshop and discussed study findings with study participants and stakeholders from the District and National Health Departments [23]. We used the Objective, Reflective, Interpretational and Decisional (ORID) framework (Fig 1) to facilitate discussions during the workshop.

## Results

Participants were mainly female with a median age 47 (range: 32–65) years. Ten (45.5%) were clinic nurses, 11 (50%) had a college diploma, 9 (41)%) had completed university and had a bachelor's degree in nursing, 2 (9%) had a medical degree. In the context of this qualitative study's results, participants refer to Black and White groups. These classifications are not racial, but rather social constructs of South African historical Apartheid origin, without biological meaning.

### Healthcare provider and manager barriers

#### 1 Lack of knowledge of policies on breast and cervical cancer prevention, early detection and management.

Most facility managers and healthcare providers stated that they were unaware of the national breast cancer prevention and control policies and guidelines. These guidelines have not been mandated for implementation in routine primary care which in turn hindered and delayed early breast cancer detection.

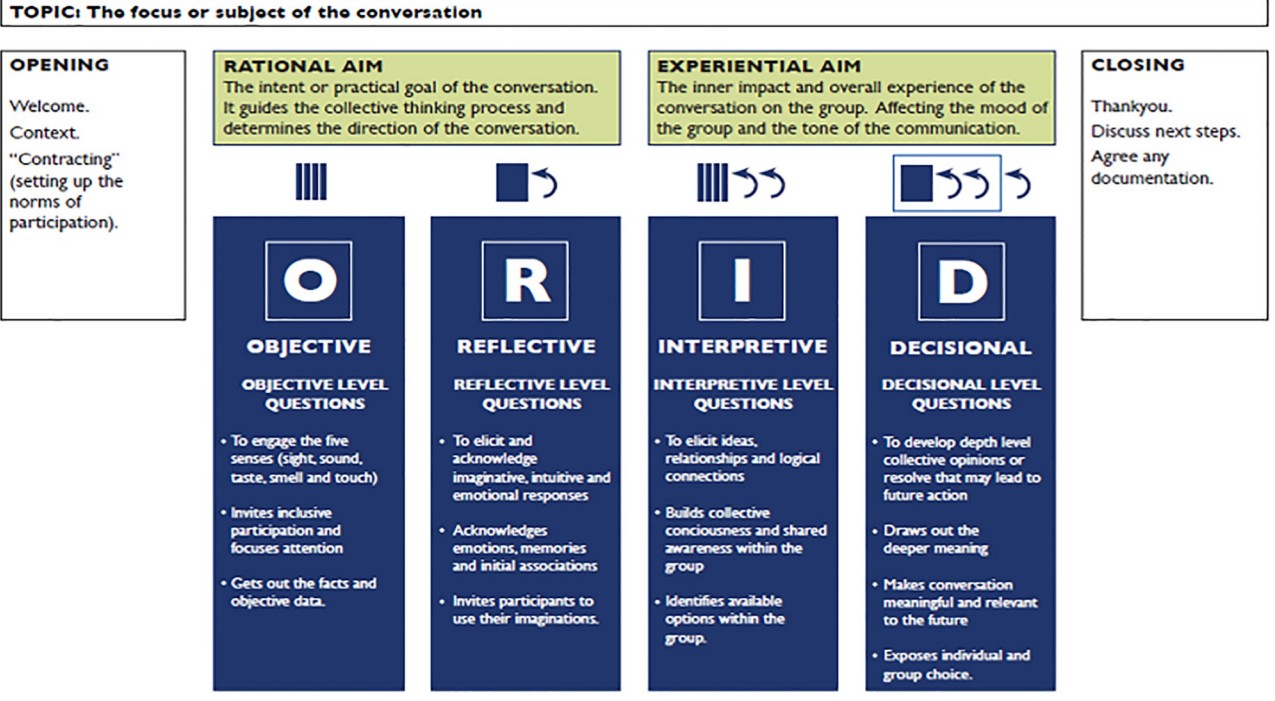

**Fig 1. ORID framework.**

*"We, as staff members, need to be empowered. Maybe if we have more knowledge. And we will have confidence to talk about breast cancer we'd be able to share what breast cancer is. How does it start? So, we also need to be knowledgeable about it so that we can be able to talk about it."*

**(IDI-014-Healthcare provider)**

*"I don't know that guideline [breast cancer screening guidelines]."*

(**IDI-002-Healthcare provider**)

### I. **Cancer is too often not recognized**

Most facility managers and healthcare providers were unsure about how to identify symptoms of cancer. Therefore, they stated often overlooking cancer symptoms and misdiagnosing the symptoms. For example, a healthcare provider reported that symptoms of cervical cancer might be mistaken for symptoms of sexually transmitted infections (STIs).

*"With a smelly discharge, we think STI first . . . you treat even if maybe it's related to the cervical cancer. But anyway, it's not harmful to treat as though it's a STI.*

(**IDI-016-Healthcare provider**)

*"Even during our training, even our guidelines they place it [cancer] last. They place everything else you can think about and lastly cancer. You would try to investigate*

*everything else then when you have failed then that is when you will say let me investigate cancer."*

(**IDI-018-Facility Manager**)

### II. Inadequate healthcare provider competency for the cancer screening procedure

Although some healthcare providers described the process of clinical breast examination and Pap smear procedure, they stated lacking the technical skills to carry out these cancer screening procedures correctly.

*"I also think us [healthcare providers], we are not equipped. We don't have information . . . we are not well trained . . . The training is needed."*

(**IDI-011-Healthcare provider**)

*"Some persons would do the Pap smear and the results would come invalid . . . you did not take the specimen correctly. So, it's a waste of time. I'm not sure how to brush I think the brush is like this (demonstrates)."*

(**IDI-001-Facility manager**)

Providers advocated for in-service training on cancer policies and guidelines and on cancer signs and symptoms identification.

*"If people can be given in service training on a regular basis to do the clinical breast examination and cervical screenings. . ."*

(**IDI-005-Facility manager**)

*"Every health care provider needs to get a copy of these guidelines, they must be legible, written and pasted in each and every consultation room. . . ... R: and every time we have a staff meeting we have to set aside some time to talk about the importance of these guidelines and the operating procedures in the clinic."*

(**IDI-015-Facility manager**)

### 2. Relying solely on patients to present at primary care clinics with self-reporting breast cancer symptoms or risks

Most participants reported that, with few exceptions, screening for breast cancer did not take place routinely in their clinics. Therefore, the onus was on patients to self-report their symptoms of breast cancer.

*So my concept is the body tells you when there's a problem. If you were to screen everybody it would take too much time. So you have to prioritize and you rely on the disease to manifest itself, with pain or lump . . . because it's just not easy all the time to screen everybody because we have to triage. Some people are more urgent, and you don't want to be wasting time on someone who's very healthy with no symptoms."*

(**IDI-016- Healthcare provider**)

Cervical cancer screening was however reported to be mandated and thus performed routinely for HIV positive woman of any age and for women at six- week postnatal visits.

*"I'll be truthful here now; our clinicians don't screen for cancer to any patient. . .. The only place where the cancer can be screened is when the patient came at family planning or at the HIV clinic because those patients are female patients, and are done a Pap smear."*

(**IDI-005-Facility manager**)

### 3. Shortages in screening facilities, supplies, and human resources

### I. No private facilities to examine patients and supply shortages

Most participants shared their frustrations with insufficient infrastructure and that shortage of cervical screening equipment delayed early cervical cancer detection.

*"And then there are glitches here and there . . . we may run out of Pap smear slides, Pap smear brushes."*

(**IDI-019-Healthcare provider**)

*"I: And then what in your experience are the barriers to early detection of cervical cancer? R: Clients who don't come early [in the day] or they came and there is no equipment. They said, there are no speculums."*

(**IDI-002-Healthcare provider**)

Sometimes healthcare providers have to share consultation rooms with their colleagues. Some participants speculated that patients might be uncomfortable to be examined in the consultation rooms because of lack of privacy.

*"The impact is huge because now we fail to perform those tests [Pap smear] on patients because we don't have those things [speculums and autoclave machine]. We don't have screens in our rooms . . . if I'm going to ask patients to take off their clothes and lie down, without this screen or without a curtain then that patient is going to feel like they're not respected. So I think that the screens will make her feel comfortable and feel that she is being respected."*

(**IDI-005-Facility Manager**)

*"Our environment doesn't allow us to do that [Pap smear]. We are two sisters in one room. We are attending two patients at the same time, privacy is not there."*

(**IDI-011-Healthcare provider**)

One facility manager provided suggestions to address the shortages in screening facilities, supplies.

*"Also provide us with resources such as your speculum's, make sure that windows have curtains and KY gel [for cervical cancer screening]."*

(**IDI-001-Facility manager**)

**II. Not enough providers for high patient volumes**

Most participants revealed that they were unable to carry out routine cancer screening on eligible woman accessing healthcare services because there were not enough healthcare providers available for the high patient volumes.

*"It's the ratio of the patients that we see. We are very few [providers] and we see a lot of patients. There are many people in the community. At times you find that you end up seeing 50 patients in a day which is impossible to see those patients if you still must do breast examination on all those patients."*

(**IDI-007- Healthcare provider**)

*"If they [government] can give us more staffing [human resource], replace those who are going on pension and increase total clinic physicians to work at the clinic and."*

(**IDI–001-Facility manager**)

**4. Staff attitude**

**III. Staff attitude towards learning new skills**

A few facility managers stated that some staff members are resistant to change and have a negative attitude towards skills development.

*"And the attitude. Everyone doesn't want to do Pap smear for whatever reason. They even do not want to do HIV testing so imagine Pap smear whereby they have to pick and there is a queue outside but then we not doing justice to our patients"*

(**IDI-017-Facility manager**)

*"You see you come into the facility, you are new, you find someone who has been working in the facility forever, for a long time but with the old knowledge and information which they don't find it easy to amend in their minds to say; oh this is what's being done now, being trained to go along with what's being done and then that also affects you who is new. Because you come in and work with me, but I am still doing things in the old ways only to find that things are being done differently. It's also resistance to change, that one is very much present in most of us.*

(**IDI-003-Facility manager**)

**I. Staff attitude towards patients**

Most participants acknowledged that the challenges for women to attend healthcare facilities were possibly related to the healthcare provider's negative attitude. They acknowledged that healthcare providers could be harsh towards patients and that in turn prevented patients from coming into the clinic or reporting with symptoms that could be suggestive of breast or cervical cancer.

*"Other people are lazy to come because of the bad attitude that we have as healthcare providers, we don't treat them with respect"*

(**IDI-002- Healthcare provider**)

*"It's very rare to find them coming in to say sister I want a Pap smear just like that, and prior to that again with us, we would be asking why patients want to do Pap smear as nurses, that's another thing that prevents patients, they don't get to be free, to say this is what I want because they will be asked questions and all that based on our attitude."*

(***IDI-003-Facility manager***)

### 5. Staff rotation which prevents mastery

Some facility managers and healthcare providers perceived staff rotation as a hindrance to cancer screening services compounded by the fact that some healthcare providers rotated into screening services were not passionate about nor dedicated to cancer screening.

*"If a person is passionate to working in a certain department and does things right. I think we need to leave that person on that post . . . Because she is passionate about Pap smears."*

(***IDI-001-Facility manager***)

## Perceived patient barriers

### 1. Healthcare providers think patients lack cancer-related knowledge

### I. Healthcare providers think that patients do not recognize cancer signs and symptoms

Most healthcare providers stated that they have observed that patients have insufficient knowledge on breast and cervical cancer signs and symptoms, resulting in late-stage presentation and diagnosis. These participants advocated for existing ward-based healthcare workers to educate women in their communities.

*"From the patient side, maybe they don't have knowledge about the signs of breast cancer. Or if they are from a family history of cancer, they don't know that this could mean cancer for them. They don't know that they must go to the clinic and that they can do self-breast examination or come to the clinic for clinical breast, or they can do a mammogram and they don't even take it seriously"*

(***IDI-015-Healthcare provider***)

*"They will sit at home with that knowledge that they would only come into the clinic when they are experiencing pain and we also need to inform them that cancer is a silent killer and that by the time they experienced pain, it could be that the cancer is at stage 2 or stage 3. But if we come early, we can detect it early so that you can get help quickly. So, it is important that the community has the information"*

(***IDI-008-Healthcare provide***)

*"I think it's health education especially because we have warbots [ward-based community workers] who do outreach in the community. Empower the warbots and then they can educate the community about the signs and symptoms to enable early detection so that they can educate the community on what to do when they notice signs and symptoms."*

(***IDI-007- Healthcare provider***)

### II. Healthcare providers think patients perceive cancer as being associated with white people

Some facility managers and healthcare providers reported that the community perceived cancer to be a "white man's" disease.

*"Some of them [patients] have this thing that this disease [cancer] is for White people and not for us Black people. . ..."*

(**IDI-003-Facility manager**)

*"We used to believe that they are just for whites, so people do not associate with it, they feel like they can never get cancer because they are not white."*

(**IDI-009- Healthcare provider**)

### III. Healthcare providers believe patients have misconceptions about cervical cancer screening procedure

Most participants reported that there is a general perception within communities that the Pap smear procedure is painful and as a result women are reluctant to undergo a Pap smear procedure.

*"R: They [patients] think that doing the Pap smear is painful, but in fact it is not painful. There is just a bit of discomfort."*

(**IDI-008- Healthcare provider**)

*"R: I think there's a perception that Pap smears are painful. This is whether from and experience from a friend hearing from a friend or it's just a myth that is just out there."*

(**IDI0-019- Healthcare provider**)

### 2. Delays in cancer screening and/or diagnosis

### I. Patients self-medicating

Healthcare providers reported that some patients first self-medicate and/or use alternative medication. Patients thus delay and only come into the healthcare facility in most cases when the cancer disease is at the advanced stage.

*"R: They procrastinate, or they go and seek help elsewhere or they treat themselves at home with whatever spirit [spiritual healing], whatever they think will work. . . .. It is traditional until it is proven otherwise."*

(**IDI-015- Healthcare provider**)

*"R: I still go back to saying that first they try our traditional herbs and that's what delays them, only when they see that now things are getting worse that's when they come to the clinic."*

(**IDI-009-Healthcare provider**)

**II. Delay in early cancer detection and management due to patient hopping from one clinic to another**

A few healthcare providers raised the concern of patients "hopping" from one healthcare facility to another as this pattern of behavior delays early cancer detection. They did not suggest any explanations for this behavior.

*"So sometimes a person will come and says, just that there are people when they do, they like doing uh what clinic shopping or medical shopping, whereby a person, a person will come to Orlando [clinic], then go to Meadowlands, then Mandela Sisulu and Lillian Ngoyi for the same problem, then go to Mandela and go to Lilian Ngoyi so there is no continuation. R: So, when it comes to me, it's a new person. **I: Hmm**. R: I don't know the history here."*

(**IDI-006-Facility manager**)

**3. Healthcare providers think patients view healthcare services as curative and not preventative.**

A few participants explained that the patients are not knowledgeable about the preventative services offered at primary healthcare facilities and that patients mostly come to healthcare facilities only when they are sick.

*"I: Outside of just the breast cancer in general, do you think our patients are aware that they can come into our clinic just for screening tests not only. R: I don't think they know. And I think also, unfortunately, because clinics are so overwhelmed in most places, they will send patients back [i.e. away] unless you have symptoms."*

(**IDI-016-Healthcare provider**)

*"So people feel that, you know, once it's when they're not sick, why test, I'm not sick?"*

(**IDI-006- Facility manager**)

## Results of the stakeholder dissemination workshop discussions

These results refer to the healthcare providers' views of patients and/or women and do not necessarily reflect the views nor behaviours of patients/women.

## Objective: Take home message from the workshop

Participants were concerned about conflicting messaging on the criteria used for cervical cancer screening, citing how some healthcare providers perform cervical cancer screening on woman presenting with cervical symptoms while other providers would treat woman with the same symptoms as sexually transmitted infection. Participants suggested ongoing in-service training for healthcare providers and community cancer outreach campaigns and individual education of ward based primary healthcare outreach team (warbots) while they are out in the community tracking and tracing patients.

## Reflective: What feelings did listening to the findings evoke in workshop attendees

Participants expressed their concern that the department of health prioritises HIV/TB policies with little emphasis on cancer policies. They also felt inadequately supported by the

department of health, citing burnout due to the high patient volumes and inadequate human and infrastructure resources. Further, they felt the healthcare system is failing healthcare providers and patients because healthcare providers are not well informed nor trained in identifying cancer signs and symptoms and management but are expected to provide routine screening services. The inadequate facilities and supplies for cancer screening was also noted as a concern by participants. Participants expressed concerns on how healthcare providers are often not involved in the policy and guideline development process and often these policies are not practical at grassroot level and also not filtered down to healthcare providers.

### Interpretive: What are the reasons for these findings, based on their lived experiences (including currently in the COVID era)

Participants shared how COVID-19 negatively impacted on the number of people that are screened for cancer. The only policy that most healthcare providers are knowledgeable about is cervical screening policy, however it is limited to screening HIV positive woman and for women at the 6 weeks postnatal visit. Participants noted that there was lack of communication and coordination among different departments within the department of health.

### Decisional: Their insight on what to do and proposed resolution to issues identified

Delegates reiterated the need for additional resources, ongoing cancer education for providers and communities, for primary care providers to participate in guidelines and implementation tools development and for better screening facilities.

Additionally, providers perceived delays in receiving cytology laboratory results as contributing to late -stage cancer diagnosis and lost to follow up of patients. There should be a functional system that allows for healthcare workers to access results as soon as they become available and to highlight outstanding results. Providers proposed for the department of health to partner with NGO's that specialize in cancer screening and management to bolster resources for breast and cervical cancer screening. They also perceived that the referral process of patients from primary healthcare to tertiary institutions often demotivates and delays in early cancer detection and management, emphasizing a need for prompt referral systems for patients to be adopted and implemented.

## Discussion

In summary, providers reported their inadequate cancer and policy knowledge and screening skills; insufficient provider resources, facilities, and supplies to accommodate required screening volumes; negative staff attitudes towards patients and to knowledge and skills acquisition, exacerbated by regular staff rotations, and lack of access to screening laboratory results. These barriers were compounded by a perceived lack of support and focus on cancer management from the National DOH. The consequences were low screening volumes and poor quality of screening services provided and a de-emphasis of preventative medicine services in primary healthcare clinics overrun by high patient volumes. Providers reported that they perceived patients to have poor cancer risk and symptoms knowledge exacerbated by perceptions of low cancer risk and screening procedures perceived as being painful. Providers believed that women prioritised self-medication and traditional healing, utilising health services only when they felt very ill. Potential solutions proffered were ongoing in-service knowledge and skills training; guideline clarification and ward- based community fieldworkers to educate communities, increased provider resources partnering with cancer NGO's, better screening facilities

and more supplies; involving providers in screening policy and guidelines development and better access to screening laboratory test results.

Our findings are aligned to recent SSA review articles by McFarland et al. [24] and Pierz et al. [25] who also revealed that women feared cancer and did not understand that screening is a preventative measure. Such findings were echoed by other researchers in Durban, SA [26] and in Western Kenya [27] who reported that although women have heard of breast and cervical cancer, their knowledge of risks, signs and symptoms of these cancers is poor, with limited awareness on the need for cancer screening. The very real perception that women have that the Pap smear process as painful, is a serious commonly reported barrier to cervical cancer screening. Patient cultural and religious barriers were also mentioned, with provider barriers including a failure to inform or encourage women to screen. Pierz et al. [25] highlighted patient perceptions of cancer risk and opinion of influential community members, trust in the health services, political will and, support and encouragement of the patient by providers, as facilitators to screening uptake. Poor primary care provider cancer knowledge was also confirmed by Heena et al in Saudi Arabia [28], and by Bateman et al. [29] Tanzania. In a recent study in the Western Cape of SA, Moodley et al. [30] highlighted the difficulty for primary care providers to distinguish symptoms of infectious diseases from those of cervical cancer in settings of high infectious disease prevalence. They pointed to a need to support primary care health care providers in assessing symptomatic patients. Inadequate resources for primary care cancer screening services were also reported by Bateman et al. [29] where clinicians revealed huge challenges in navigating large patient volumes in crowded clinics, faulty equipment, and unreliable power supplies and by Diala PC et al in Western Kenya [27].

We sought to explore the complex interactions of our findings to highlight areas for further research and to reveal potential opportunities for intervention. Utilising the socioecological [31] and capability, opportunity and motivation model of behaviour (COM-B) models, we firstly grouped our findings and explored pathways that influence low screening provision [32]. The COM-B model identifies what needs to change for a behavioural intervention to be effective. The COM-B model identifies three factors that need to be present for any behaviour to occur: capability, opportunity and motivation. As depicted in Fig 2, the SA Department of Health is perceived not to provide sufficient cancer training support and there is insufficient trained staff as they are regularly rotated to other clinic services. Providers thus lack sufficient knowledge and skills on cancer, screening policies and techniques. Patients and communities who also have poor cancer knowledge, perceive Black South Africans not to be susceptible to cancer. There is thus currently low capacity for cancer screening services in primary care clinics and low community uptake and demand. From an opportunity perspective, breast cancer screening is currently not mandated by the SA DOH and the limited cervical cancer screening services provided are compromised by insufficient providers, inadequate facilities, supplies and barriers to accessing screening laboratory results. Women in turn prefer to self-medicate and consult with traditional healers, regarding clinics as for curative not preventative services, hopping from clinic to clinic, all of which serve to delay access. The result is low opportunity to provide and demand cancer screening services. And because the National SA Health Department is perceived not to prioritize cancer and does not involve primary car stakeholders in policy and performance indicator development, there is little motivation for providers to offer screening services. This is reflected in overworked staff having little appetite to learning cancer screening skills, being unwelcoming to patients who in turn prefer to go elsewhere and perceive cervical cancer screening as painful.

Our results highlight the need to explore these determinants and others such as economic constraints and rationale for staff rotations among policy makers, stigma, self-efficacy, and preferences for preventative services among women in clinics and communities. Our results

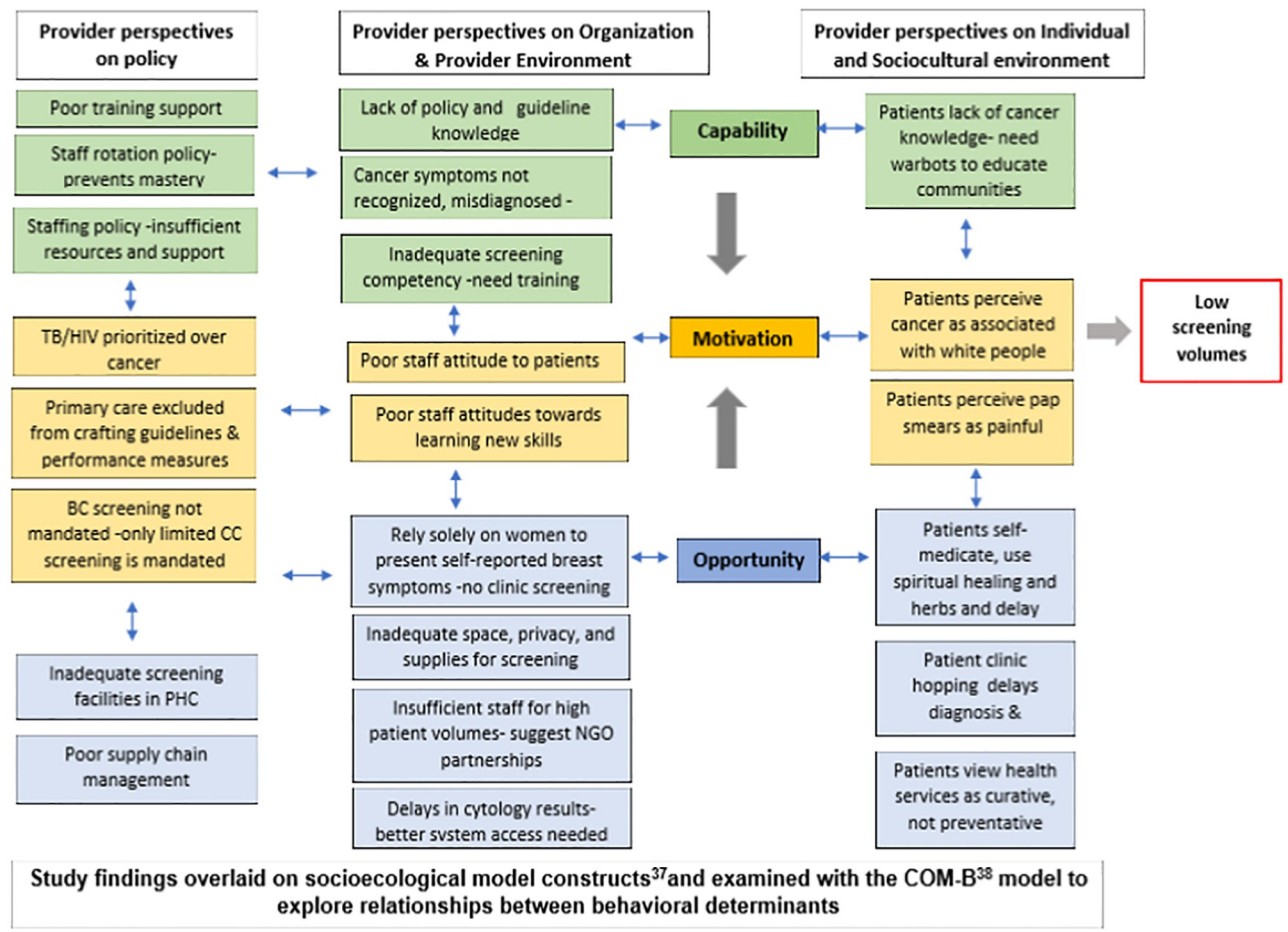

**Fig 2. Summary of study findings and their complex interactions -provide perspectives.**

also reveal opportunities to address the low screening uptake. The existing community worker, cancer NGO sector and traditional healer workforce with training may be able to alleviate provider shortages. The lack of screening facilities can be addressed with relatively inexpensive screening structures such as mobile clinics and tents; central and clinic supply chain management systems can be better implemented, providers can be given access to National laboratory services electronic screening results system and inexpensive IT tools can be developed to alert for outstanding screening results and patient visits, all with multistakeholder involvement.

## Strengths and limitations of the study

A few participants indicated that were nervous when they were invited to participate in an interview about the detection and management of cancer. They were under the impression that this would serve as an evaluation on their knowledge and work relating to cancer. The Interviewer; GT who is an experienced qualitative research interviewer with a background in social worker managed to create a conducive environment by firstly indicating to the participants she is not an expert in the topic as well and will in the process also be learning. These concerns were addressed before and during the consenting process. After the discussion, the participants were still eager and interested to participant in the in-depth interview.

Our study provides an in-depth investigation of provider barriers and patient challenges for screening uptake for SA women in primary health clinics. Although the context was limited to peri-urban Greater Soweto, Johannesburg, our findings were confirmed by those from various other SA and SSA country settings. This study did not however include sufficient medical officers (I.*e*., general practitioners) from the representative clinics, so the sample in the healthcare providers was overrepresented by nurses. Having said that, the SA Public Health System is predominantly nurse driven and solutions will have to be tailored to nurse provider and ultimately patient and community needs. The results of this study present provider experiences and perceptions of patients and/or women. Patients/women may not necessarily hold these views or behave in the way referred to by providers.

## Conclusion

Our findings reveal primary healthcare provider perspectives about the complex health policy, organization, provider and potential patient factors that interact in compounding ways to negatively impact cancer screening service provision and community uptake in Greater Soweto communities. Encouragingly they also reveal opportunities to address the barriers. These determinants need verification and further exploration among multiple stakeholder groups to understand how best to increase provision (supply) and demand for cancer screening services. Our findings highlight the need for further engagement and research among community, patient, and provider stakeholder groups to further understand determinants and opportunities to increase screening rates.

## Supporting information

**S1 Table. Coding tree: Highlights the coding process and data analyses process.**
(DOCX)

**S2 Table. ISSM COREQ checklist.**
(DOCX)

## Acknowledgments

We are particularly grateful to all the study participants for the time and information they shared with us and the support received from the Gauteng National and Johannesburg district department of health for their shared support.

## Author Contributions

**Conceptualization:** Maureen Joffe, Janan Janine Dietrich.

**Data curation:** Gugulethu Tshabalala, Charmaine Blanchard, Keletso Mmoledi, Desiree Malope, Maureen Joffe, Janan Janine Dietrich.

**Formal analysis:** Gugulethu Tshabalala, Janan Janine Dietrich.

**Funding acquisition:** Maureen Joffe.

**Investigation:** Gugulethu Tshabalala, Maureen Joffe, Janan Janine Dietrich.

**Methodology:** Maureen Joffe, Janan Janine Dietrich.

**Project administration:** Maureen Joffe, Janan Janine Dietrich.

**Resources:** Maureen Joffe, Janan Janine Dietrich.

**Supervision:** Janan Janine Dietrich.

**Writing – original draft:** Gugulethu Tshabalala, Maureen Joffe, Janan Janine Dietrich.

**Writing – review & editing:** Gugulethu Tshabalala, Charmaine Blanchard, Keletso Mmoledi, Desiree Malope, Daniel S. O'Neil, Shane A. Norris, Maureen Joffe, Janan Janine Dietrich.

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
