## [Decision Letter · Decision Letter 0]

7 Dec 2022

PGPH-D-22-01548

A qualitative study to explore healthcare providers’ perspectives on barriers and enablers to early detection of breast and cervical cancers among women attending primary healthcare clinics in Johannesburg, South Africa

Dear Dr. Dietrich,

Thank you for submitting your manuscript to PLOS Global Public Health. After careful consideration, we feel that it has merit but does not fully meet PLOS Global Public Health’s publication criteria as it currently stands. Therefore, we invite you to submit a revised version of the manuscript that addresses the points raised during the review process.

We look forward to receiving your revised manuscript.

Kind regards,

Jennifer Moodley

Academic Editor

Journal Requirements:

a. State what role the funders took in the study. If the funders had no role in your study, please state: “The funders had no role in study design, data collection and analysis, decision to publish, or preparation of the manuscript.”

b. If any authors received a salary from any of your funders, please state which authors and which funders.

2. Since your data is not available for proprietary reasons, please explain via email why the data is not available. Please also include the contact information for the third party organization that should be contacted should other researchers want to request access to this data and please include the full citation of where the data can be found. We also request that you verify with us via email that any researcher will be able to obtain the data set in the same manner that the you have obtained it. If you feel you are unwilling or unable to adhere to this policy, please explain your reasons by return email and your exemption request will be escalated to the editor for approval. Your exemption request will be handled independently and will not hold up the peer review process, but will need to be resolved should your manuscript be accepted for publication. One of the Editorial team will be in touch if they require more information.

Additional Editor Comments (if provided):

In addition to addressing the Reviewers comments, could the authors please also address the following:

1. Line 90 “ bare” should be “bear”

2. The authors refer to Black and White patients throughout the study. Use of such terms can lead to unintentional bias. Could the authors please justify use of this terminology, making clear the origins of these terms and that these racial classifications are social constructs without biological meaning.

3. The use of the term “pap” throughout the article needs to be revised. The correct abbreviation is “Pap” Could the authors please use the full term the first time it is used.

4. In the Western Cape study referred to in Line 606 providers did consider cancer as a differential diagnosis but highlighted the difficulty of distinguishing between symptoms of infectious diseases and cervical cancer at a primary care level which is important to consider given the high burden of STIs in the setting. It pointed to a need to support primary care providers in assessing symptomatic patients.

Reviewers' comments:

Reviewer's Responses to Questions

**Comments to the Author**

1. Does this manuscript meet PLOS Global Public Health’s publication criteria? Is the manuscript technically sound, and do the data support the conclusions? The manuscript must describe methodologically and ethically rigorous research with conclusions that are appropriately drawn based on the data presented.

Reviewer #1: Yes

Reviewer #2: Partly

2. Has the statistical analysis been performed appropriately and rigorously?

Reviewer #1: N/A

Reviewer #2: N/A

3. Have the authors made all data underlying the findings in their manuscript fully available (please refer to the Data Availability Statement at the start of the manuscript PDF file)?

Reviewer #1: Yes

Reviewer #2: No

4. Is the manuscript presented in an intelligible fashion and written in standard English?

Reviewer #1: Yes

Reviewer #2: Yes

5. Review Comments to the Author

Reviewer #1: This is an important study with a clear rationale and detailed account of methods. The study provides useful, insightful and sometimes concerning insights into barriers to early cancer detection. The authors have embedded findings in existing model of behaviour and this works well.

There are a few issues that would benefit from consideration:

Make sure the text is clear that the findings refer to the providers’ views of patients/women (e.g. lines 587, 624, 619). Women may not necessarily hold these views/behave this way.

Similarly, suggest headings in Figure 2 are ‘Providers perspectives of policy’, ‘Providers perspectives of organisation and staff’ ‘Providers perspective of individual and socio-cultural issues’

The section on ‘enablers for early breast and cervical cancer detection’ felt quite repetitive as it highlighted opposites of the previously described barriers (e.g. barrier = lack of training; enabler = training opportunities). Suggest this section is merged with the previous section.

Figure 2 is useful but would benefit from better formatting to align boxes etc

The reference section needs attention. Check the references are numbered in the order they appear in the text and numbers correspond to the correct reference in the list. Also check all are in the list: references 20-23 and 36-38 appear missing from the list.

Other minor points:

Line 547 ‘burnout’ not ‘burn out’

Spell out COM-B acronym and give brief overview of this model

Check if LMIC refers to low-to-middle income countries or low and middle income countries.

Reviewer #2: Thank you for the opportunity to review this manuscript, which uses data from qualitative interviews with health providers and clinic managers in Soweto, South Africa to report on factors at multiple levels – policy, organisational, and individual (providers and users) – that may explain low breast and cervical cancer screening volumes at primary care clinics in Soweto, South Africa, and possible solutions for addressing the identified barriers. I commend the authors for addressing these relevant questions because despite enabling policies, breast examination and cervical screening and pre-cancer treatment coverage remain low in South Africa, and solutions are needed. The manuscript has some strengths and potential to contribute to the literature. It also has weaknesses that need addressing before it is fit to publish. Below are suggestions for improving the quality of the manuscript.

As an overall comment, it is problematic that the manuscript draws conclusions about patient-side barriers based on provider perceptions. This approach does not adequately represent user / patient voices in the health system. The manuscript also draws some conclusions that are not supported by the data and seem to be over-reach. The manuscript does however report some useful findings, but its main message must be articulated better and coherence between the problem, aim and results must be improved if it is to appeal to a broader South African and international audience.

I have uploaded the review as an attachment

6. PLOS authors have the option to publish the peer review history of their article (what does this mean?). If published, this will include your full peer review and any attached files.

**Do you want your identity to be public for this peer review?** For information about this choice, including consent withdrawal, please see our Privacy Policy.

Reviewer #1: **Yes: **Suzanne Scott

Reviewer #2: No

---

## [Decision Letter · Decision Letter 1]

5 Feb 2023

PGPH-D-22-01548R1

A qualitative study to explore healthcare providers’ perspectives on barriers and enablers to early detection of breast and cervical cancers among women attending primary healthcare clinics in Johannesburg, South Africa

Dear Dr. Dietrich,

Thank you for submitting your revised manuscript to PLOS Global Public Health. After careful consideration, we feel that it has merit but does not fully meet PLOS Global Public Health’s publication criteria as it currently stands. Therefore, we invite you to submit a revised version of the manuscript that addresses the points raised during the review process.

EDITOR: 

Please address the issues raised by Reviewer 1. These relate to some headings and statements that do not adequately indicate that the views are that of heath care providers rather than patients. For this qualitative study, please complete the COREQ checklist.

We look forward to receiving your revised manuscript.

Kind regards,

Jennifer Moodley

Academic Editor

Journal Requirements:

Additional Editor Comments (if provided):

Please could the authors address the issues raised by Reviewer 1.

In addition for this qualitative study, please complete the COREQ checklist.

Reviewers' comments:

Reviewer's Responses to Questions

**Comments to the Author**

1. If the authors have adequately addressed your comments raised in a previous round of review and you feel that this manuscript is now acceptable for publication, you may indicate that here to bypass the “Comments to the Author” section, enter your conflict of interest statement in the “Confidential to Editor” section, and submit your "Accept" recommendation.

Reviewer #1: (No Response)

Reviewer #2: All comments have been addressed

2. Does this manuscript meet PLOS Global Public Health’s publication criteria? Is the manuscript technically sound, and do the data support the conclusions? The manuscript must describe methodologically and ethically rigorous research with conclusions that are appropriately drawn based on the data presented.

Reviewer #1: Yes

Reviewer #2: Partly

3. Has the statistical analysis been performed appropriately and rigorously?

Reviewer #1: N/A

Reviewer #2: N/A

4. Have the authors made all data underlying the findings in their manuscript fully available (please refer to the Data Availability Statement at the start of the manuscript PDF file)?

Reviewer #1: No

Reviewer #2: No

5. Is the manuscript presented in an intelligible fashion and written in standard English?

Reviewer #1: Yes

Reviewer #2: Yes

6. Review Comments to the Author

Reviewer #1: On the whole the reviewers’ comments have been addressed. However the results section is still a little misleading as the headings do not reflect that these are healthcare providers views (rather than patients). Suggest these are altered e.g. “Healthcare providers think patients lack cancer related knowledge”; “Healthcare provides think patients do not recognise cancer signs and symptoms”; “Healthcare providers think patients perceive cancer as being associated with white people”; “Healthcare providers believe patients have misconceptions about cervical cancer screening procedures”; “Healthcare providers think patients view healthcare services as curative not preventative”.

This caution should be applied in the discussion too, particularly lines 572 – 575.

Reviewer #2: None

7. PLOS authors have the option to publish the peer review history of their article (what does this mean?). If published, this will include your full peer review and any attached files.

**Do you want your identity to be public for this peer review?** For information about this choice, including consent withdrawal, please see our Privacy Policy.

Reviewer #1: **Yes: **Suzanne Scott

Reviewer #2: No

---

## [Editor Report · Decision Letter 2]

23 Mar 2023

A qualitative study to explore healthcare providers’ perspectives on barriers and enablers to early detection of breast and cervical cancers among women attending primary healthcare clinics in Johannesburg, South Africa

PGPH-D-22-01548R2

Dear Prof Dietrich,

We are pleased to inform you that your manuscript 'A qualitative study to explore healthcare providers’ perspectives on barriers and enablers to early detection of breast and cervical cancers among women attending primary healthcare clinics in Johannesburg, South Africa' has been provisionally accepted for publication in PLOS Global Public Health.

Best regards,

Jennifer Moodley

Academic Editor